Quantifying inter-group variability in lactation curve shape and magnitude with the MilkBot® lactation model

Ehrlich James L. jehrlich@dairyvets.com
Dairy Veterinarians Group , Argyle, NY , USA
Uversky Vladimir
Electronic publication date: 2013 Mar 12
Publication date: 2013
Volume: 1
Electronic Location ID: e54
Received 2013 Jan 22; Accepted 2013 Mar 1
Copyright: © 2013 Ehrlich
Copyright year: 2013
Copyright holder: Ehrlich
License: This is an open access article distributed under the terms of the Creative Commons Attribution License, which permits unrestricted use, distribution, and reproduction in any medium, provided the original author and source are credited.
License URL: https://creativecommons.org/licenses/by/3.0/

Keywords: Lactation curve, Persistency, MilkBot, Dairy management, Lactation

Funding: USDA National Institute for Food and Agriculture (NIFA) SBIR grant 2008-33610-18962 Data used in this study was acquired from Dairy Records Management Systems (Raleigh, NC, and Ames, IA) under SBIR grant 2008-33610-18962 from the USDA National Institute for Food and Agriculture (NIFA). The funders had no role in study design, data collection and analysis, decision to publish, or preparation of the manuscript.

==============================
Genetic selection programs have driven development of most lactation models, to estimate the magnitude of animals’ productive capacity from sampled milk production data. There has been less attention to management and research applications, where it may also be important to quantify the shape of lactation curves, and predict future daily milk production for incomplete lactations since residuals between predicted and actual daily production can be used to quantify the response to an intervention. A model may decrease the confounding effects of lactation stage, parity, breed, and possibly other factors depending on how the model is constructed and used, thus increasing the power of statistical analyses. Models with a mechanistic derivation may allow direct inference about biology from fitted production data. The MilkBot® lactation model is derived from abstract suppositions about growth of udder capacity. This permits inference about shape of the lactation curve directly from parameter values, but not direct conclusions about physiology. Individual parameters relate to the overall scale of the lactation, the ramp, or rate of growth around parturition, decay describing the senescence of productive capacity (inversely related to persistence), and the relatively insignificant time offset between calving and the physiological start of milk secretion. A proprietary algorithm was used to fit monthly test data from two parity groups in 21 randomly selected herds, and results displayed in box-and-whisker charts and Z-test tables. Fitted curves are constrained by the MilkBot® equation to a single peak that blends into an exponential decline in late lactation. This is seen as an abstraction of productive capacity, with actual daily production higher or lower due to random error plus short-term environmental effects. The four MilkBot® parameters, and metrics calculated directly from them including fitting error, peak milk and cumulative production, can be used to describe and compare individual lactations or groups of lactations. There is considerable intra-herd and inter-herd variability in scale, ramp, decay, RMSE, peak milk, and cumulative production, suggesting that management and environment have significant influence on both shape and magnitude of normal lactation curves.

Background

Lactation models typically predict milk production, or a milk component, as a function of DIM (days in-milk). They may be classified by several criteria that have important implications for their usefulness in applications. First, models may be classified as mechanistic or empirical based on their derivation and objectives. For example Dijkstra et al. (1997) and Pollott (2000) each describe a mechanistic model developed from data and assumptions about cell populations in mammary tissue, from which inferences can be made about milk production. Because of this derivation, data about milk production can also lead back to inferences about physiology through the construct of the model. In contrast, empirical models such as Best Prediction (Cole & VanRaden, 2006) and many other systems developed for estimating genetic value of bulls from the records of their daughters are content with predicting milk production without any hypothetical linkage to the biology of lactation. The seminal Wood model (Wood, 1967) was introduced without a mechanistic derivation, though the portion of the model controlling senescence (identical to the corresponding part of the MilkBot® model and several other models) can be interpreted mechanistically.

For both mechanistic and empirical models it is often desirable to fit production data to the model generating a set of fitted parameter values describing a lactation curve. For linear models, or models like Best Prediction, which fits a linear adjustment to a nonlinear base, linear regression yields a simple and reliable fitting method. Some widely used polynomial models also are easily fitted (Schaeffer et al., 2000), but may present other difficulties. Fitting methods are less straightforward for most other nonlinear functions. For models that may be transformed mathematically to a linearized form, like the Wood model, linear regression can still be used, but the solution is not very satisfactory because the transformation of raw data means the result is no longer a true least-squares fit. Many techniques exist for true least-squares optimization of nonlinear models, but matching a model and fitting method in a way that generates consistently reliable results is complicated by characteristics of both model and data. Despite this difficulty, increased computing power, and improvements in nonlinear fitting methods, have made nonlinear models a practical alternative for fitting both individual and aggregated lactations. Fitting of nonlinear models often involves iterative algorithms for which it cannot be guaranteed that solutions are unique, optimal, or even findable. This means that extensive and varied testing is required to evaluate the robustness of a nonlinear model-and-fitting-method pair when applied to data sets of various kinds. A model which accurately reflects the mechanism behind the data is often more robust because the mathematics of the model naturally constrains solutions to those which are biologically probable.

The question of what constitutes an optimal fit deserves more attention than it sometimes receives. The usual assumption is that the best fit minimizes the sum of squares of residuals, often referred to as mean square error (MSE), or its square root, RMSE (root mean square error). It can be argued that this solution is most likely to be correct if error is normally distributed and there is no prior information about the population being measured. In reality, we have a great deal of information on what normal lactations look like, and how they vary with factors such as breed, parity, region, and so on. Maximum likelihood methods incorporate some of this prior information on expected frequency of solutions along with the test data during optimization, to arrive at solutions that are more likely to reflect reality, despite MSE that is normally higher than obtainable by minimizing MSE strictly, especially if there are few data points.

Data to be fitted may be from a single lactation, or aggregated data from some grouping of multiple lactations in which case the curve should be called an aggregate lactation curve. Aggregated data is influenced by population effects in addition to individual lactation effects so that studies on aggregated data are not completely comparable to studies based on fitting of individual lactations. For example management policies for culling usually depend on relative production of individuals, so that higher merit individuals may be over-represented in late lactation. This could cause an upward deflection of aggregate data in late lactation, as is sometimes seen (Ehrlich, 2011; Dematawewa, Pearson & VanRaden, 2007), without a matching deflection in individual lactations. Many such hidden population effects are possible, so it should not be assumed that the model that best fits aggregated data also fits individual lactations best. Aggregation of data also may either decrease or increase MSE, depending on whether aggregation is done before or after fitting. For example, Dematawewa, Pearson & VanRaden (2007) fitted single curves to data points from multiple lactations, which would be expected to increase MSE relative to individual lactations because of added variability between lactations. Ehrlich (2011) fitted a large data set aggregated by DIM before fitting, which decreased MSE to very low levels. Either of these approaches can be supported, but the meaning of MSE values is altered radically.

Applications of lactation models vary widely. Much work has been done with the objective of obtaining an estimate of the genetic value of milking animals and their sires, notably at the USDA Animal Improvement Programs Laboratory (USDA-AIPL) (Animal Improvement Programs: Home, 2012) in Beltsville, MD and by other members of the International Bull Evaluation Service, “Interbull” (2012). For this application, models may adjust for effects of lactation stage, breed, parity, season, and region on production, and environmental effects are seen as confounding variables, while the main point of interest is an individual animal’s productive capacity. For management purposes, though, it is often the environmental effects that are the variables of interest. For example a dairyman may want to know the effect of a change in transition management or a researcher the effect of a feed additive. Changes in the distribution of milk within lactations, i.e. the shape of the curve, may be more sensitive and specific in measuring effects of short-term changes such as illness, feeding, or housing, but these can only be quantified if model parameters relate to curve shape in a consistent way. This was demonstrated by Hostens et al. (2012) in a study of relations between metabolic diseases and milk production. Looking at raw milk production data or M305 (cumulative milk production for the first 305 days of a lactation), no statistically significant relations were found, but there were multiple significant or highly significant relations between diseases and parameter values for individual lactations fitted to the MilkBot® model. Transition period diseases tended to hurt the ramp of the lactation, but there was a compensatory increase in persistency leading to little net change in M305. Through the model, these effects on the distribution of production within the lactation could be quantified as changes to fitted parameter values of individual lactations.

A model fitted to individual lactations sometimes can be used to predict daily milk production, and study the effect of management and environment on milk production through analysis of residuals. Residuals, that is the difference between expected and observed daily milk production, can be attributed to factors outside the model domain such as feeding, illness, the environment, or some management intervention. Model residuals minimize the confounding effect of DIM on production, without the smoothing effect of choosing a cumulative metric like M305.

The MilkBot® model has been used with the Levenburg-Marquardt fitting algorithm to fit aggregate lactation curves for major breeds and parities (Ehrlich, 2011). The proprietary DairySight genetic algorithm is also available online (Ehrlich, 2012) to fit individual lactations to the MilkBot® model. This uses a maximum likelihood algorithm to return a solution even with no data points, in which case population mean parameter values will be returned. As more data points become available, they are combined with the a priori information, and individual solutions will diverge from population means. This means that sensitivity and specificity of the fit to real differences between lactations will increase as more data points are used. While the process obviously cannot detect differences in parts of the curve where there are no data points, a stable solution is achieved in nearly all cases, and solutions are usually insensitive to small changes to data points. Cole, Ehrlich & Null (2012), Hostens et al. (2012) and Charlier et al. (2012) used the DairySight fitting engine successfully in fitting individual lactations from monthly test data.

No model is perfect, nor is testing ever complete. Relative performance of models may depend on details of the population studied and what data is used. Normal production curves may vary by breed, parity, and other groupings, and may change over time, so testing should be continuous and in multiple forms. One basic test is to make appropriately blinded predictions based on the model and tabulate statistics on prediction error. Cole, Ehrlich & Null (2012) showed that MilkBot® predications were usually more accurate and precise than three other lactation models by calculating mean and standard deviation of error in predicting next-test milk as a function of DIM. A more general test is simply to see whether application of a model leads to useful results. In this way, hundreds of papers on genetic selection of cattle using lactation models, and the dramatic improvements in milk production of dairy cattle they facilitated testify to the utility of AIPL and Interbull models. Similarly, recent papers by Hostens et al. (2012) and Charlier et al. (2012) testify to the utility of the MilkBot® model in measuring management effects.

That management, breed, and parity influence magnitude of production is obvious, and can be measured by M305, peak milk, or similar metrics, but little is known about variability in lactation curve shape, except that first-parity animals typically show greater persistency and lower peak milk than older cows. It is often assumed that curve shape is biologically fixed, and that management influences magnitude only, but this assumption may be a consequence the difficulty of quantifying shape. Quantification of lactation curve shape, or the distribution of production within a lactation, is hampered by lack of a standard methodology and terminology. For example there is no standard quantitative definition of “persistency” in general usage, and no generally accepted quantitative metric of the rate of rise in early lactation that within the MilkBot® model is called ramp.

Development of the MilkBot® lactation model was driven by the belief that management of dairy herds influences the shape of lactation curves, and that those differences can be measured statistically as a means of studying effects of management and environment on milk production. The null hypothesis, in that context, is that lactation curve shape does not vary significantly between dairy herds and parity groups, or that if such variability exists the MilkBot® model provides no benefit in detecting and quantifying that variability. An important secondary objective is to provide background information for experimental design and the building of specific hypotheses about effects of management and environment on lactation curve shape. Because no data was available on management of individual groups and herds, such causal relationships are entirely speculative in this study.

Methods

Development of the MilkBot® model

Rook, France & Dhanoa (1993) describe a general lactation model consisting of a growth process multiplied by a death process and a scalar. They stipulate that both growth and death processes should be monotonic, with the growth function rising from zero to approach one, and the death function decreasing from one to approach zero. These constraints have the important effect of making the scalar largely responsible for magnitude of the curve with the growth function dominating shape of the rising portion of a normal lactation curve, and the death function dominating the decline in late lactation, or decay portion of the curve. Note that the growth function returns the cumulative growth as a function of time, not the rate of growth. This may be inferred from the constraints. The multiplicative variant of the Pollot mechanistic model  (Pollott, 2000) follows this general model, but the frequently cited Wood (1967) lactation model, while otherwise similar to the general model, has no upper limit to the growth function, which results in the scaling of the lactation curve being shared between the scalar and the growth process.

In developing the MilkBot® model, we accept the constraints of the Rook general model and begin by extending it backwards, defining a Growthrate Function, C(t) which models the rate of creation of lactational capacity. This means the Growth function will be the integral of the Growthrate function, and Growthrate will be the derivative of Growth. Equation (1) states the general model as defined, and extended, where Y(t) represents milk production on day t of a lactation with scalar a and G(t), D(t), and C(t) representing growth, death, and Growthrate functions. (1) Y(t)=aG(t)D(t)=a∫−∞tC(t)dtD(t).

To maintain conformity with standard practices, the model’s independent variable, t, represents DIM, with t = 1 at parturition, but in a further extension of the general model, and contrary to current standard practice, we note that parturition and the start of lactation are separate events, though related. We define the start of lactation as the midpoint of growth in production capacity, and an offset variable, c, which is the time between parturition and that growth midpoint (the start of lactation). Therefore G(t) = 1/2 when t = c by definition.

Though C(t) is not strictly a probability distribution, it is similarly constrained, with the full-range integral equal to 1. The most prominent probability distribution, the normal distribution, is an obvious candidate for C(t), which would require the rate of creation of udder capacity to rise in a bell curve to a peak near parturition, and then fall again. This has the advantage of being at least moderately credible in relation to observable growth of udder tissue, and easily described by two familiar parameters corresponding to the standard deviation of the Normal curve and the offset location parameter.

The integral of the Normal distribution, the Cumulative Normal function, can only be calculated as an improper integral, which adds considerable mathematical complication. Therefore, simply as a concession to ease of computation, MilkBot® uses an approximation, the Laplace distribution, for C(t), because it is easily integrated, while yielding a curve of similar shape. The Laplace distribution takes different mathematical forms for left and right sides of the curve, but since the peak is expected to be at or near parturition, and we are unconcerned with the shape of growth before milking commences, the left side of the function can safely be dropped (Eq. (2)). This is equivalent mathematically to postulating that growth rate of udder capacity decreases exponentially from a peak that defines the start of lactation, and with enough high quality data it might be possible to discern whether an exponential or Gaussian growth rate function is closer to reality, but differences between the two are very small after the two weeks of lactation. The resulting growth function (Eq. (3)) ends up mathematically equivalent to the Mitscherlich function described by Rook, France & Dhanoa (1993), but with a different parameterization which emphasizes similarity with the initial Gaussian model. (2) Ct=12bexpc−tbwheret>c(Laplace distribution)

(3) Gt=∫−∞tC(t)dt=1−expc−tb2.

For a Death Function, D(t), we choose the same exponential decay function used by Wood, Rook, and others. Dematawewa, Pearson & VanRaden (2007) examined models for extended lactations and found that models featuring an exponential decline of this form outperformed other candidates for lactations exceeding 305 days in length. (4) Dt=exp(−dt).

Substitution of Eqs. (3) and (4) in Eq. (1) yields the MilkBot® Model. (5) Yt=a1−expc−tb2exp(−dt)(MilkBot® model).

Experimental data

Groups of 50 lactations each were drawn from 21 randomly selected dairy herds, and then compared to see whether herd effects influence lactation curve shape. Herds were selected from a large DHIA database of herds predominantly in the eastern half of the USA and containing monthly milk weights from more than six million lactations in over 17,000 herds. This is the same database used earlier for developing standard breed-parity aggregate curves (Ehrlich, 2011). Initial selection chose 1,056 herds with at least 1,000 recorded lactations between January 2005 and June 2008. From these, 21 herds were selected randomly. All data points after 305 DIM were dropped, then all lactations with fewer than 7 recorded monthly test days rejected. Lactations were divided in two parity groups, with all lactations after the first in a single group. Each group was then ordered by calving date and one lactation chosen randomly as a starting point. Lactations were then selected forward and backwards in time until 50 lactations had been collected. If a herd could not provide sufficient qualified lactations in either group, that herd was replaced by another randomly selected herd. This resulted in a set of 21 herds, each with 2 parity groups of 50 complete lactations having sequential calving dates. Lactations were all fitted using the DairySight fitting engine (Ehrlich, 2012), and parameter values recorded. Data for each herd-parity group was recorded as comma-delimited data in a single text file, then imported into Mathematica,1 which was used to generate tables and figures. The number of herds chosen was set to 21 because using the cutoff p < .05 in statistical testing leads to a 1/20 chance of falsely rejecting the null hypothesis. Therefore each herd could be compared to 20 others, with the expectation that on the average there would be one false-positive at p < .05 for each parameter compared. Greater differences suggest greater-than-random variability between herds.

Results and Discussion

The conceptual derivation described under Methods leads to the MilkBot® model, giving daily milk yield, Y(t) as a function of DIM (t) as shown in Eq. (1). Y(t)=a1−expc−tb2exp(−dt).

Model parameters a, b, c, and d control shape and magnitude of the lactation curve. These parameters are given descriptive names based on the functions from which they were derived and their effect on the general model.

Interpretation of parameters

Parameter “a” is the scale parameter. It can be expressed as kilograms/day, pounds/day, or similarly. Scale can be seen as the theoretical maximum daily yield, which approaches actual peak production as ramp approaches zero. It must be a positive number. Changing the model to a different unit of measure for milk output only requires applying the appropriate conversion to the scale parameter, while all other parameter values remain unchanged. Scale can be seen as describing the magnitude of a lactation, while the other three parameters describe the curve’s shape.

Parameter “b” is the ramp parameter, controlling the width of the Growthrate function, and so the rate of rise in milk production in early lactation. Ramp values are time, normally days, and must be a positive number. A simple thumb rule is that when t = r a m p, growth in production capacity is about 82% complete.

Parameter “c” is the offset parameter describing the offset in time between parturition and the start of lactation. Offset values are time (days), and may be positive, negative, or zero. The effect of offset is slight, except in the first few days of the lactation curve, so it is generally not possible to detect variation in the offset parameter in fitted lactations unless there are daily milk weights covering the first weeks of lactation. With a monthly interval between test points, values for the offset parameter have little value, and the model may be simplified by setting offset equal to zero or a constant near zero. Fixing offset to zero is equivalent to the common assumption that lactation begins at parturition.

Parameter “d” is the decay parameter, controlling the rate of senescence of production capacity. Decay is inverse-time (days−1). It should be constrained to positive values under normal circumstances. An equivalent alternative expression for a first-order decay constant like the decay parameter is as a half-life, which suggests a definition of lactation persistency, which, rather than being tied to an arbitrary stage of lactation, is an attribute of the lactation as a whole. (6) Persistence=0.693decay.

By this definition decay is the time it would take for production to drop by half, if we were to ignore the growth side of the model. Since the Growth function approaches one in late lactation, decay, by this definition, is close to the actual half-life for milk production in late lactation. Fitted decay values are likely to be more normally distributed than persistence, therefore decay should be preferred for most statistical calculations, but may be converted to persistence afterwards.

Mathematical manipulation of Eq. (1) allows calculation of some useful results. By setting the derivative equal to zero we can calculate peak day, tpeak, and from that peak milk, ypeak. Note that like persistence, tpeak and ypeak are attributes of the lactation as a whole, and less sensitive to any single data point than metrics based on comparing individual test days. (7) tpeak=c−b log[(2bd)/(1+bd)]

(8) ypeak=Ytpeak=aexp(−d(c−b Log[2])[a,b,c,d])(1−1/2exp((c−(c−b Log[2])[a,b,c,d])/b)).

Cumulative production between two days can be calculated by integration, including M305. (9) M305=(a−aexp(−305d))/d+(abexp(c/b)(−1+exp(−305(1/b+d))))/(2+2bd).

Comparison of herd-parity groups

Herd-parity groups were compared to establish whether MilkBot® parameters or derived metrics (RMSE, M305, peak day, peak milk) could be used to detect statistical differences in lactation curves between groups. It can be inferred that if such differences exist between herds, they probably are caused by differences in management or environment, and this suggests that statistical analysis of fitted parameter values or derived metrics may be a means of monitoring herd performance and detecting abnormalities. Existing literature firmly establishes that decay should be lower in first-parity groups (higher persistence in heifers), and that there are likely to be differences in measures of magnitude such as scale, peak milk, and M305 between herds and parities. It is much less certain whether management and environment influence persistency (decay), ramp, offset, RMSE, and time of peak milk, or whether these might be fixed biologically, and insensitive to management.

Table 1 gives herd mean M305, predominant breed, and links to files containing data on 50 individual lactations for each of the 42 groups. Table 2 summarizes data for all herd-parity groups for each parameter or derived metric. These are overall means of the group mean and group standard deviation, and mean divergence score. A “divergence score” was calculated for each group by tabulating how many of the 20 other groups of matching parity had significantly different mean parameter value by Z-test at P < .05, so that 1.0 would be the expected divergence score if there are not differences among groups, and higher scores suggest divergence is more common than would be expected from random variation.

Table 1 Herd data.

Mean M305, predominate breed, and data files for 21 randomly selected dairy herds (Supplemental Data files are available at: Supplemental Information).

	Mean M305 (kg)	Breed	50 parity 1 lactations	50 parity 2 +lactations	
Herd A	6284	JERSEY	Supplemental Data (file 1.csv)	Supplemental Data (file 2.csv)	
Herd B	7822	CROSSBRED	Supplemental Data (file 3.csv)	Supplemental Data (file 4.csv)	
Herd C	8005	HOLSTEIN	Supplemental Data (file 5.csv)	Supplemental Data (file 6.csv)	
Herd D	8741	HOLSTEIN	Supplemental Data (file 7.csv)	Supplemental Data (file 8.csv)	
Herd E	9621	HOLSTEIN	Supplemental Data (file 9.csv)	Supplemental Data (file 10.csv)	
Herd F	9730	HOLSTEIN	Supplemental Data (file 11.csv)	Supplemental Data (file 12.csv)	
Herd G	9796	HOLSTEIN	Supplemental Data (file 13.csv)	Supplemental Data (file 14.csv)	
Herd H	10037	HOLSTEIN	Supplemental Data (file 15.csv)	Supplemental Data (file 16.csv)	
Herd I	10338	HOLSTEIN	Supplemental Data (file 17.csv)	Supplemental Data (file 18.csv)	
Herd J	10345	HOLSTEIN	Supplemental Data (file 19.csv)	Supplemental Data (file 20.csv)	
Herd K	10481	JERSEY	Supplemental Data (file 21.csv)	Supplemental Data (file 22.csv)	
Herd L	10580	HOLSTEIN	Supplemental Data (file 23.csv)	Supplemental Data (file 24.csv)	
Herd M	10909	HOLSTEIN	Supplemental Data (file 25.csv)	Supplemental Data (file 26.csv)	
Herd N	11118	HOLSTEIN	Supplemental Data (file 27.csv)	Supplemental Data (file 28.csv)	
Herd O	11189	HOLSTEIN	Supplemental Data (file 29.csv)	Supplemental Data (file 30.csv)	
Herd P	11285	HOLSTEIN	Supplemental Data (file 31.csv)	Supplemental Data (file 32.csv)	
Herd Q	11332	HOLSTEIN	Supplemental Data (file 33.csv)	Supplemental Data (file 34.csv)	
Herd R	11521	HOLSTEIN	Supplemental Data (file 35.csv)	Supplemental Data (file 36.csv)	
Herd S	11715	HOLSTEIN	Supplemental Data (file 37.csv)	Supplemental Data (file 38.csv)	
Herd T	11987	HOLSTEIN	Supplemental Data (file 39.csv)	Supplemental Data (file 40.csv)	
Herd U	12370	HOLSTEIN	Supplemental Data (file 41.csv)	Supplemental Data (file 42.csv)	

Table 2 Mean MilkBot® parameter statistics for 2 parity groups in 21 randomly selected dairy herds.

Means for all herds of group-mean parameter values and group standard deviation, and divergence score. Divergence score is mean number of matching parity groups (of 20 possible) with which individual groups differ by Z-test at P < .05 (Supplemental Data files are available at: Supplemental Information).

Metric and parity	Mean group mean	Mean group sd	Mean divergence score	
Scale P1 (kg)	38.66	5.87	13.9	
Scale P2+(kg)	53.65	9.3	14.7	
Ramp P1 (days)	31.43	2.67	10.8	
Ramp P2+(days)	26.13	7.66	8.8	
Offset P1 (days)	−0.5	0	0	
Offset P2+(days)	−0.37	0.39	4.6	
Decay P1 (day−1)	0.000974	0.000605	11.3	
Decay P2+(day−1)	0.002213	0.000858	12.5	
RMSE P1 (kg)	3.73	1.51	12.5	
RMSE P2+(kg)	4.65	2.04	11.1	
M305 P1 (kg)	9637	1460	14.6	
M305 P2+(kg)	11255	1811	16	
PeakDay P1	89.83	32	7.2	
PeakDay P2+	59.18	15.87	12.8	
PeakMilk P1 (kg)	33.87	6.18	13	
PeakMilk P2+(kg)	44.86	7.06	15.5	
Data file P1	Supplemental Data (file 43.csv)	Supplemental Data (file 44.csv)	Supplemental Data (file 45.csv)	
Data file P2+	Supplemental Data (file 46.csv)	Supplemental Data (file 47.csv)	Supplemental Data (file 48.csv)	

Figure 1 Distribution of cumulative 305-day milk production (M305) for herd-parity groups of 50 consecutive lactations in 21 randomly selected herds.

Herds are ordered by herd average M305 with highest at the top. Diamonds show 95% confidence around mean by Z-test. Colored bars show quartiles. Whiskers show full range.

Figure 2 Distribution of fitted MilkBot®scale parameter for herd-parity groups of 50 consecutive lactations in 21 randomly selected herds.

Herds are ordered by herd average M305 with highest at the top. Diamonds show 95% confidence around mean by Z-test. Colored bars show quartiles. Whiskers show full range.

Figure 3 Distribution of fitted MilkBot®ramp parameter for herd-parity groups of 50 consecutive lactations in 21 randomly selected herds.

Herds are ordered by herd average M305 with highest at the top. Diamonds show 95% confidence around mean by Z-test. Colored bars show quartiles. Whiskers show full range.

Figure 4 Distribution of fitted MilkBot®offset parameter for herd-parity groups of 50 consecutive lactations in 21 randomly selected herds.

Herds are ordered by herd average M305 with highest at the top. Diamonds show 95% confidence around mean by Z-test. Colored bars show quartiles. Whiskers show full range.

Figure 5 Distribution of fitted MilkBot®decay parameter for herd-parity groups of 50 consecutive lactations in 21 randomly selected herds.

Herds are ordered by herd average M305 with highest at the top. Diamonds show 95% confidence around mean by Z-test. Colored bars show quartiles. Whiskers show full range.

Figure 6 Distribution of root-mean-square fitting error (RMSE) for herd-parity groups of 50 consecutive lactations in 21 randomly selected herds.

Herds are ordered by herd average M305 with highest at the top. Diamonds show 95% confidence around mean by Z-test. Colored bars show quartiles. Whiskers show full range.

Figure 7 Distribution of day of peak milk production (peak day) for herd-parity groups of 50 consecutive lactations in 21 randomly selected herds.

Herds are ordered by herd average M305 with highest at the top. Diamonds show 95% confidence around mean by Z-test. Colored bars show quartiles. Whiskers show full range.

Figure 8 Distribution of peak milk production (peak milk) for herd-parity groups of 50 consecutive lactations in 21 randomly selected herds.

Herds are ordered by herd average M305 with highest at the top. Diamonds show 95% confidence around mean by Z-test. Colored bars show quartiles. Whiskers show full range.

Figures 1–8 show box-and-whiskers plots for M305, scale, ramp, offset, decay, RMSE, peak day, and peak milk. In each plot, herds are ordered by herd-average M305, with the lowest-production herd (Herd A) at the bottom. Parity groups are differentiated by color. For each group, a gray diamond gives the Z-test 95% confidence interval around the mean. Colored bars show quartiles above and below the mean, and whiskers show the full range for the group. This means that groups with diamonds that overlap vertically are not significantly different by Z-test with 95% confidence. If diamonds do not overlap, the sample means differ at p < .05. Groups as small as 50 lactations with monthly test data are clearly sufficient to identify significant variability among herds, with the exception of offset and possibly peak day. Scale, M305, and peak milk seem to follow similar patterns, suggesting the correlation that is to be expected, as all are primarily measures of lactation magnitude.

Ramp is higher for heifers, indicating a slower rise in production after calving. It can be speculated that inter-herd variability in ramp might be influenced by transition management. Some herds (C, I, L, N, R, T) show little difference in ramp between parity groups, and since transition heifers are often managed differently from cows, it is easy to develop hypotheses on possible causes. For example, the differences could relate to whether fresh heifers are group-housed with older animals since competition with older animals might limit feed consumption in heifers, and slow growth in production capacity. That the same herds tend to show a wide range in ramp values among mature cows adds another angle for speculation and research.

There is little difference in offset within or between herds, which is not surprising since only monthly milk weights were available and biologically significant differences in offset would have little effect past the first few days of lactation. Even so, in mature cows there are statistically significant differences in mean offset of as much as 0.3 days (8 h). It is very unlikely that this reflects biologically or economically significant differences in offset. Rather, with only monthly data points the fitting engine has difficulty differentiating offset from ramp, so makes the appropriate choice of attributing most but not all variability to ramp. This leads to correlation between offset and ramp as an artifact of the fitting process. One simple solution is to ignore offset when monthly test data is used. The untested possibility remains that true differences in offset would be found with daily milk weights.

Heifers have markedly lower decay (greater persistency) than mature cows, but there is also considerable variation between herds. For example, persistency of mature cows in Herd K is slightly better than heifers in Herd C. The difference between cows and heifers also varies widely among herds. Factors that might influence persistency include use of rBST (recombinant bovine somatotropin), nutrition, mastitis control, and many other variables in management, environment, and genetics. These could have considerable economic importance through the effect on overall production.

Distribution of RMSE is interesting because greater consistency either in the testing protocol generating production data or in actual production should lower RMSE. Mastitis, lameness, poor feed management, and many other problems would likely increase RMSE so that it may be useful as a general, if imperfect, measure of management quality and cow welfare. These are extremely difficult to measure by objective criteria, so even imperfect methodology may be useful. For example, Herd T has high production but also relatively high RMSE, which might reflect management problems, or just lower quality data collection. There are a few outliers, such as individual lactations in herds N and T with very high RMSE. For example herd N includes a single lactation with RMSE of 24.8 kg and milk weights recorded as (47.7, 83.1 84.4, 32.2, 8.2, 55.4, 77.2, 64.5, 34, 24.1) kg at approximately monthly intervals. This is obviously a highly abnormal lactation, and probably a cow with health problems. Some portion of RMSE should be attributed to bias inherent in the model design and fitting process, since no model or fitting process is perfect. This systemic error should be randomly distributed among herds, so the inter-herd variability in RMSE suggests at least that systemic error is not overwhelming. The remaining portion is sometimes called random error, reflecting the high variability that is expected even between consecutive milkings. This probably is a misnomer, since most of the variability probably has a cause, whether it is inconsistency in milking times, variability in feeds, meter error, weather, disease, or any of thousands of other possible causes. Clearly some of these are controllable, and some are not, but RMSE allows them to be measured, at least collectively.

Peak milk and peak day will correlate with the parameters from which they are calculated. It is possible that they may turn out to correlate better with certain variables of interest than individual parameters, but that remains to be seen. Peak milk shows a pattern similar to scale, but with more difference between cows and heifers because of the lower decay which is characteristic of heifers. Peak day is not greatly influenced by production level, and shows relatively little variation among herds, suggesting that it is more fixed biologically, and mostly insensitive to management and environment.

Peak milk is sometimes used with the thumb rule that each unit of peak milk translates to 250 units in M305, or some similar formula. This presupposes that lactation curve shape is the same in all herds, and is not influenced by whatever management changes might improve peak milk. We can get a feel for the practical effect of herd variability in curve shape by looking at its effect on this thumb rule. A rather complicated formula2 can be derived to calculate how much change there would be in M305 if scale were increased enough to increase peak milk by one unit. This depends on values for the other three parameters, especially decay. For Herd C, for example, a unit of peak translates to 230 units of M305 for heifers, and 188 for cows while in Herd Q these are 267 and 221 units respectively.

Conclusions

The semi-mechanistic derivation of the MilkBot® model assigns clear meanings to the parameter scale, as a measure of magnitude and ramp, offset, and decay parameters as measures of lactation curve shape. The decay parameter also suggests a definition of persistency expressed as a half-life called persistence that is an attribute of the lactation as a whole rather than any particular portion. Parameters may easily be used to calculate M305, peak milk, peak day, expected production at any day in the curve, or cumulative production for a portion of a lactation.

The null hypothesis is rejected. Lactation curve shape does vary between herd-parity groups, in multiple ways. Those differences can be quantified statistically by fitting individual lactations to the MilkBot® model and summarizing fitted parameter values. At least some of these differences may be supposed due to management differences between groups, but little is known about possible causes.

The semi-mechanistic derivation of the MilkBot® model may also suggest hypotheses that particular interventions might influence particular parameters, or metrics calculated from them such as M305, peak milk, or peak day. Fitting field or experimental data to the model allows such hypotheses to be tested. This methodology appears to be capable of quantifying effects that would be difficult or impossible to detect using only raw milk weights or M305. Similarly, parameters can be used to predict future milk in incomplete lactations. Then residuals can be summarized as a measure of the effect of short-term changes in management or environment.

Nonlinear models are in many ways more difficult to work with than linear models, but may represent biology better than is possible when models are limited to linear or polynomial functions. Techniques for fitting observed data to nonlinear models vary greatly, and performance of a fitting engine often depends on characteristics of the data set being fitted as well as the model. Despite the difficulties, nonlinear models can facilitate insight into complex processes of biology, and the MilkBot® model has been shown both to predict future milk production in individual lactations with good accuracy and to be capable of quantifying what appear to be biologically and economically significant differences in lactation curve shape between groups of animals.

Milk production is a very sensitive indicator of cow health, as well as being the most important economic driver for dairy farms. The DairySight fitting engine (Ehrlich, 2012) can successfully fit individual lactations from monthly test data, though significant RMSE is to be expected, returning a set of parameter values that describe the lactation as a whole. RMSE of fitted lactations can be used as a measure of consistency. By quantifying shape, magnitude, and consistency of lactations in a repeatable manner it is hoped that much future work will be enabled.

Supplemental Information

Supplemental Information 1 Supplemental comma-delimited data

Individual files as described in Table 1

Click here for additional data file.

Additional Information and Declarations

Competing Interests

Author Contributions

1 Mathematica version 8.0.1.0 from Wolfram Research.

2 PeakToM305Multiplier = (2 − 2 Eˆ(−305 decay) +2 decay ramp − 2 decay Eˆ(−305 decay) ramp +decay Eˆ(−305 decay − 305/ramp +offset/ramp) ramp − decay Eˆ(offset/ramp) ramp)/(2 (decay +decayˆ2 ramp)).

I am a private-practice veterinarian serving dairy herds near Argyle, NY. Some work related to MilkBot was funded by a 2008 grant from USDA under their SBIR program, including a payment to DRMS for data used here and elsewhere. I have permission to publish that data, which has been anonymized. The SBIR program requires that proposals have prospects for commercialization. To that end I filed a MilkBot patent application and registered US trademarks, but I have officially abandoned the patent application and now provide free MilkBot services on the Internet. I have vague hopes of possible future income as a consultant, or in licensing software, but I have no income related to this project now or in the foreseeable future, and the MilkBot model is firmly in the public domain.

James L. Ehrlich conceived and designed the experiments, performed the experiments, analyzed the data, contributed reagents/materials/analysis tools, wrote the paper.

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
