# Peer review of "Quantifying inter-group variability in lactation curve shape and magnitude with the MilkBot® lactation model"

_PeerJ, doi:10.7717/peerj.54_

## Round 0.1 · original submission · Major Revisions

Please pay close attention to the comments of reviewer #1 who made several deficiencies in your manuscript.

·

Basic reporting

This manuscript is well-written and conforms to journal format guidelines. The information presented is highly technical, but the author has done a good job in describing the information. All references are appropriately cited within the text and the citations are through relative to the scope of this topic in the literature. Unfortunately there is a flaw to the manuscript in that the author never provides an objective statement for the reader to appreciate what is to be undertaken in the study. It is uncertain if the author is attempting to validate his mathematical model with data, or describe the methodology by which he determined this model. The model has been previously used to assess its ability to discern critical assessment of lactational performance of cows relative to other models or standard Dairy Herd Improvement record metrics. How does this manuscript differ from the two previous publications in which the author collaborated? There are other issues within the manuscript that do detract from the clarity of the text for the reader. A number of equations are provided by the author; however, the defined number for equations is repeated within the text. Each equation should have an independent number for clarity. Duplication of the equation as in line 214 is not necessary. Additionally, parameters within the equation presented should have individual terms defined. Equation 2 within the Methods section is not cited within the text. Variables used in equations are not defined until the Results section, which again brings some uncertainty as to the study objectives. Legends for the figures require additional information for the reader to understand the abbreviations used. Figure legends should stand on their own and not require the reader to refer back to the text to understand.

Experimental design

Without an objective statement being provided it is difficult to assess the study design. If the intent of the study was to use the data to define parameters for the MilkBot model, the discussion is reasonably appropriate. However, if the intent of the study was to show herd differences or applicability of the MilkBot model, then the results are somewhat flawed. The description of the experimental data defines two parity groups, which suggests the author has already determined that there are no differences between parity 2 and 3 or greater in the model's parameters, unlike what is calculated in DHIA data. This is fine, but the author should show statistical assessment to validate this assessment. Also, in reviewing all the herd data, why was there no statistical assessment of model parameters relative to herd contribution to variation. It would seem that herd and parity would be two model main effects that could be tested relative to influence on a, b, c, and d parameters. In contrast, the author just describes such differences.

Validity of the findings

There is no statistical methods used in the study to verify study findings. The study was not truly testing any hypothesis, but describing a methodology for assessing lactational performance. As a result, most of the discussion is speculation by the author on how the model is describing differences in production curves. Overall validity of findings is obscured due to the lack of defined direction for the study.

Additional comments

Deficiencies noted with the manuscript can easily be corrected by providing an objective statement for the study and then basing descriptions of methods on this stated hypothesis. Statistical analysis of the different model parameters (table 2) by parity and herd could easily address some of concerns raised if the objective was to describe a new mathematical model to assess lactational performance in cows.

Reviewer 2 ·

Basic reporting

No Comments

Experimental design

No Comments

Validity of the findings

No Comments

Additional comments

This is a very interesting study on quantifying lactation curve shape and magnitude with MilkBot lactation model. Dr. Ehrlich presents that the MilkBot model has been shown both to predict future milk production in individual lactations with good accuracy and to be capable of quantifying what appear to be biologically and economically significant differences in lactation curve shape between groups of animals. This is a big progress to quantify shape, magnitude, and consistency of lactations in a repeatable manner. I would like to recommend it for publication as is.

---

## Round 0.2 · accepted · Accept

Therefore, you definitely did a very good job in answering critical points. Thank you again for the excellent submission. I hope that you will consider PeerJ for your future manuscripts.